# The Use of Pre-Endoscopic Metoclopramide Does Not Prevent the Need for Repeat Endoscopy: A U.S. Based Retrospective Cohort Study

**DOI:** 10.3390/life14040526

**Published:** 2024-04-19

**Authors:** Mark Ayoub, Carol Faris, Julton Tomanguillo, Nadeem Anwar, Harleen Chela, Ebubekir Daglilar

**Affiliations:** 1Department of Internal Medicine, Charleston Area Medical Center, West Virginia University, Charleston, WV 25304, USA; 2Department of General Surgery, Marshall University, Huntington, WV 25755, USA; farisc@marshall.edu; 3Division of Gastroenterology and Hepatology, Charleston Area Medical Center, West Virginia University, Charleston, WV 25304, USA

**Keywords:** PUD, EGD, gastrointestinal bleed, erythromycin, metoclopramide, prokinetic

## Abstract

Background: Peptic ulcer disease (PUD) can cause upper gastrointestinal bleeding (UGIB), often needing esophagogastroduodenoscopy (EGD). Second-look endoscopies verify resolution, but cost concerns prompt research on metoclopramide’s efficacy compared to erythromycin. Methods: We analyzed the Diamond Network of TriNetX Research database, dividing UGIB patients with PUD undergoing EGD into three groups: metoclopramide, erythromycin, and no medication. Using 1:1 propensity score matching, we compared repeat EGD, post-EGD transfusion, and mortality within one month in two study arms. Results: Out of 97,040 patients, 11.5% received metoclopramide, 3.9% received erythromycin, and 84.6% received no medication. Comparing metoclopramide to no medication showed no significant difference in repeat EGD (10.1% vs. 9.7%, *p* = 0.34), transfusion (0.78% vs. 0.86%, *p* = 0.5), or mortality (1.08% vs. 1.08%, *p* = 0.95). However, metoclopramide had a higher repeat EGD rate compared to erythromycin (9.4% vs. 7.5%, *p* = 0.003), with no significant difference in transfusion or mortality. Conclusions: The need to repeat EGD was not decreased with pre-EGD use of metoclopramide. If a prokinetic agent is to be used prior to EGD, erythromycin shows superior reduction in the need of repeat EGD as compared to metoclopramide.

## 1. Introduction

Peptic ulcer disease (PUD) continues to be a leading cause of upper gastrointestinal bleeding (UGIB), necessitating urgent intervention such as upper endoscopy [1]. While second-look endoscopies have historically been employed to confirm bleeding resolution, concerns regarding increased procedural risks and healthcare costs have prompted a search for strategies to minimize the need for repeat procedures. Prokinetic agents like erythromycin and metoclopramide have been investigated for their potential to enhance endoscopic visualization and reduce the requirement for repeat endoscopies during hospitalization [2,3,4]. However, there remains a gap in knowledge regarding the benefit of metoclopramide and its optimal use in clinical practice, leading to uncertainty in guideline recommendations.

Prokinetic agents play a crucial role in enhancing gastric emptying and improving endoscopic views, which could aid in more accurate diagnosis and potentially reduce the need for repeat procedures in patients with UGIB. Prokinetics act on multiple neurotransmitters, which exert different actions depending on the receptor of attachment [5,6,7]. In 2012, the American Society for Gastrointestinal Endoscopy (ASGE) suggested the possibility of using prokinetic agents like erythromycin before performing endoscopy on a case-by-case basis [2,3]. Similarly, they recommended the administration of erythromycin before the procedure to improve diagnostic accuracy, though it was noted that such a practice did not necessarily change patient outcomes. By 2015, the European Society of Gastrointestinal Endoscopy (ESGE) was more definitive in their guidance, advocating strongly for pre-endoscopy erythromycin use. 

The scope of our study is focused on evaluating the benefit of pre-endoscopy administration of metoclopramide in patients with UGIB. Our study focused on comparing metoclopramide to erythromycin and placebo when administered prior to EGD in patients with UGIB. It also focuses on evaluating the need for repeat EGD as a surrogate for a second-look endoscopy to ensure UGIB control. The stance of the above guidelines is supported by research, including our own findings, which demonstrate erythromycin’s effectiveness over placebo and its superiority to metoclopramide in minimizing the need for a follow-up endoscopy. By assessing the benefits and potential drawbacks of these agents, this research aims to provide valuable insights to inform clinical decision-making and optimize patient care protocols for managing UGIB.

## 2. Materials and Methods

### 2.1. Statistical Analysis

This is a retrospective cohort study that was approved by the Institution Board Review Committee at Charleston Area Medical Center on 9 May 2023 with the following IRB number: 23-956. Written informed consent from patients was waived due to the de-identified nature of the TriNetX clinical database. The TriNetX (Cambridge, MA, USA) database is a global federal research network that combines real-time data with electronic medical records. Our study was conducted using the TriNetX database through the Diamond Network, which comprises 92 Healthcare Organizations (HCO). Patients with UGIB with PUD undergoing EGD were identified. PUD was defined as acute or chronic ulceration with hemorrhage or perforation. Patients with UGIB with PUD undergoing EGD were identified using the codes from International Classification of Diseases (ICD)-10 and Current Procedural Terminology (CPT) codes. A full list of all codes used for the study as well as a full description of study definition and variables is highlighted in the Appendix A.

Descriptive statistics were utilized to outline the study population and summarize demographic characteristics, presenting data as mean (range) or frequency (percentage). To ensure comparability, patients across groups were matched using propensity score matching 1:1 matching algorithm via the TriNetX Analytics platform. Statistical significance in baseline characteristics was determined using χ^2^ tests for categorical variables and independent *t*-tests for continuous variables. Effect sizes and 95% confidence intervals (Cis) were reported to denote the magnitude of differences and precision of estimates. Hazard ratios, odds ratios (Ors), and their respective 95% Cis were calculated using TriNetX Analytics. The Kaplan–Meier method and hazard ratio (95% CI) were used to evaluate the differential impact of treatment modalities on overall survival. Data analysis was conducted between December 2022 and January 2023 on the TriNetX platform.

### 2.2. Inclusion and Exclusion Criteria 

Patients with UGIB with PUD undergoing EGD were identified and then divided into three cohorts; the first cohort included patients receiving metoclopramide, the second cohort included patients receiving erythromycin, and the third cohort included patients not receiving either medication. Two study arms were created; the first arm compared metoclopramide to no medications, and the second arm compared metoclopramide to erythromycin. Comparison was performed using 1:1 propensity score matching (PSM) based on baseline patient characteristics. A full list of variables used in PSM with their corresponding codes is highlighted in the Appendix A. We compared the need for repeat EGD, post-EGD transfusion and mortality rate in 1-month post-EGD in both study arms. The study flow is depicted in Figure 1.

## 3. Results

### 3.1. Baseline Characteristics

A cohort consisting of 97,040 patients diagnosed with upper gastrointestinal bleeding (UGIB) associated with peptic ulcer disease (PUD) and undergoing esophagogastroduodenoscopy (EGD) was identified for this study. Among these patients, 11.5% (*n* = 11,161) received metoclopramide and 3.9% (*n* = 3753) received erythromycin, while the majority, comprising 84.6% (*n* = 82,126), did not receive any medications. Following propensity score matching (PSM), analysis of baseline demographics and comorbidities across the cohorts revealed no statistically significant differences.

The mean age of patients administered metoclopramide was 56.7 years, with a standard deviation (SD) of 16. Nearly half of the patients receiving metoclopramide were female, accounting for 46.6% of the cohort. Among this group, 17% had heart failure, and 27% had ischemic heart disease. Approximately 8% of patients in this cohort required blood transfusion. Detailed comparisons of baseline demographics before and after PSM are provided In Table 1 and Table 2.

### 3.2. Outcomes

After PSM, we compared different outcomes between two cohorts in each study arm over a 1 month timeframe. 

The first study arm compared patients with UGIB with PUD undergoing EGD who received metoclopramide to those who did not receive metoclopramide or erythromycin.

Our analysis did not show any statistically significant difference in the need to repeat EGD between patients receiving metoclopramide compared to those who did not receive any medications (10.1% vs. 9.7%, *p* = 0.34). Moreover, there was no statistically significant difference in the need for post-EGD blood transfusion (0.78% vs. 0.86%, *p* = 0.5) or mortality (1.08% vs. 1.08%, *p* = 0.95). A graph plotting comparative outcomes between metoclopramide compared to no medications is shown in Figure 2.

The second study arm compared patients with UGIB with PUD undergoing EGD who received metoclopramide to those who received erythromycin.

Our analysis showed that patients receiving metoclopramide had a higher rate of repeat EGD compared to those receiving erythromycin (9.4% vs. 7.5%, *p* = 0.003). The need for blood transfusion (0.8% vs. 0.5%, *p* = 0.19) or mortality rate (1.4% vs. 0.9%, *p* = 0.1) was not significantly different between the two cohorts. A graph plotting comparative outcomes between metoclopramide and erythromycin is shown in Figure 3.

The third study arm compared patients with UGIB with PUD undergoing EGD who received erythromycin to those who did not receive metoclopramide or erythromycin.

Our analysis showed that patients receiving erythromycin had a statistically significant lower rate of repeat EGD compared to those not receiving any medications (7.5% vs. 9.5, *p* = 0.003). They also had a lower rate of mortality (1% vs. 1.6%, *p* = 0.02) and lower rate of post-EGD blood transfusion (0.5% vs. 1%, *p* = 0.02). A graph plotting comparative outcomes between erythromycin and no medications is shown in Figure 4. A summary of our overall study findings is shown below in Table 3.

## 4. Discussion

### 4.1. Concept

Upper gastrointestinal bleeding (UGIB) poses a critical medical emergency, often necessitating urgent endoscopic intervention to mitigate associated mortality and morbidity [1]. Despite the decreasing incidence of UGIB, rebleeding still occurs in up to 16% of cases after intervention [8]. Such statistics underscore the paramount importance of achieving optimal mucosal visualization to accurately identify the bleeding source and effectively prevent rebleeding [1]. However, the presence of blood products in the stomach makes thorough mucosal visualization difficult, thereby limiting the ability to identify the bleeding source and intervene promptly.

To address this challenge, the proposition of utilizing a prokinetic agent to facilitate the transition of blood products and consequently expose the underlying gastric mucosa has been advocated. This approach aims to enhance endoscopic visualization, thereby improving the efficacy of interventions aimed at controlling bleeding and preventing rebleeding episodes. By facilitating mucosal exposure and aiding in the identification of bleeding sources, prokinetics have the potential to significantly improve patient outcomes in the management of UGIB.

### 4.2. Normal Gastric Physiology and Dynamics

Gastric motility is controlled by autonomic nerves [5,6,7]. Both parasympathetic and sympathetic motor nerves exerted excitatory and inhibitory effects on the stomach, respectively. Later, studies debunked the role of sympathetic nerves in the physiological regulation of gastric motility, while the vagus nerves exert both excitatory and inhibitory effects [9,10,11]. A variety of neurotransmitters and chemicals are involved in gastric emptying and exert different effects according to the receptor type [12].

Motilin accelerates gastric emptying and is released from the M cells in response to duodenal alkalization [13,14]. Duodenal acidification causes the release of two components: prostaglandin E2 (PGE2) and 5HT. PGE2 in turn inhibits acid secretion and initiates the alkalization process, while 5-hydroxytryptamine (5HT) binds to 5HT4 receptors to initiate the release of bicarbonate in the duodenum to further increase the alkalization of duodenum [15]. Bound 5HT4 receptors also initiate duodenal contraction, which in turn improves gastric emptying [15]. Motilin also acts on the myenteric neurons and smooth muscles, and its action on the myenteric plexus promotes gastric emptying [16]. 

Dopamine is another neurotransmitter that is released endogenously and plays a part in gastric motility. Dopamine typically has an excitatory effect via dopamine 1 (D1) or an inhibitory effect via dopamine 2 (D2) receptors. D2 receptors typically have a more profound effect compared to D1 receptors [17]. Subsequently, D2 receptor antagonists accelerate gastric emptying [18]. Motilin and dopamine are the two major neurotransmitters involved in the mechanisms of action of erythromycin and metoclopramide, respectively. Other endogenously released neurotransmitters are involved in the gastric emptying process and are targets for other therapies that should be considered in future studies.

Cholecystokinin (CCK) is a prototype released from neuroendocrine cells in the duodenum in response to various stimuli. CCK plays a role in slowing down gastric emptying via the stimulation of afferent vagal endings in the inhibitory circuit [19]. It also stimulates the non-adrenergic non-cholinergic inhibitory neurons in the myenteric plexus [20]. 

Leptin is another agent that is released from the chief cells in response to protein load and vagal stimulation. Leptin acts through CCK1 receptors; however, its role in delaying gastric emptying is mediated by its effect on the hypothalamic nuclei [21]. 

Glucagon-like peptide (GLP1) is released from the intestine and stimulates vagal afferent nerves in a similar fashion to CCK [22]. The overall effect of GLP-1 was found to be slowing of gastric emptying, a decrease in the number of forward flow pulses, the inhibition of antropyloric pressure waves, and an increase in basal pyloric tone [23,24]. 

Pancreatic hormones such as insulin and amylin were also found to slow gastric emptying [12]. Pancreatic polypeptide also stimulates the gastric inhibitory vasovagal reflex, leading to slowed gastric emptying [25].

In addition to motilin, ghrelin is another stimulator of gastric emptying. Ghrelin is released from the G cells in the stomach as well as specific neurons in the hypothalamus [26]. Ghrelin acts on many receptors centrally and peripherally to exert its effect. It inhibits the inhibitory portion of dorsal motor nucleus of the vagus (DMV), disinhibits the excitatory portion of the DMV, increases gastric electrical activity, and facilitates the action of motilin [14,27,28,29,30,31]. A brief summary is shown in Figure 5.

### 4.3. Background

Endoscopic intervention is considered a first-line diagnostic modality as well as a therapeutic option for patients who present with bleeding ulcers. The incidence of peptic ulcer disease (PUD) is decreasing; however, globally, the incidence remains at 1–3% [32]. Rebleeding is common in PUD, noted in 10–15% of cases. Routine second-look endoscopies have historically been performed in PUD to verify the resolution of bleeding; however, there is rising concern about the increased cost of care and predisposition to increased risk of adverse events from an additional procedure and associated anesthesia [32,33]. We continue investigating methods to improve the visualization of an upper G.I. blood loss source and reduce the need for additional upper gastrointestinal endoscopies (EGD) [33]. Some methods for improving initial visualization include nasogastric tube placement with gastric lavage and prokinetic agents such as erythromycin and metoclopramide. Prokinetic agents are currently utilized in several clinical scenarios, including gastroparesis and video capsule endoscopy, due to the increase in small intestinal peristaltic wave velocity and contractility that they induce [34,35,36]. They have been shown to decrease the gastric transit time, have clinical utility in the intensive care setting to aid gastric emptying, and are well-tolerated [35,36]. In our study, we focus on metoclopramide and erythromycin due to their ease of availability, as well as emerging data studying both agents.

### 4.4. Medication under Study: Erythromycin

Erythromycin is a bacteriostatic antibiotic; however, it also acts as a non-peptide motilin agonist. This provides erythromycin with its gastrointestinal motility stimulation ability and further accelerates gastric emptying [37,38,39,40]. It achieves this role via two mechanisms: first, by directly improving smooth muscle contractility; second, by stimulating cholinergic neurons, leading to peristaltic movements [37]. At the smooth muscle level, it binds to motilin receptor, which activates the G-protein-coupled pathway, which in turn enhances contraction and induces gastric emptying [37,40]. Furthermore, it has potent activity when given IV by increasing antral contraction and possible fundal contractions [37,40,41,42,43]. This mechanism is the principle of its use in upper GI bleeding. Its action of gastric emptying acceleration reduces the time for endoscopy and improves visualization [44,45].

### 4.5. Medication under Study: Metoclopramide

Metoclopramide is a benzamide with prokinetic properties. Its mechanism of action is via dopamine receptor antagonism and 5-HT4-receptor agonism in the gut [46,47]. Dopamine has a relaxant effect on the gut by activating D2 receptors in the lower esophagus, fundus and antrum of stomach muscles. Additionally, it inhibits intrinsic myenteric cholinergic neurons, which leads to musculature inhibition [48]. Metoclopramide itself enhances motility via three mechanisms: the inhibition of pre- and post-synaptic D2 receptors, stimulation of pre-synaptic 5-HT4 receptors, and inhibition of pre-synaptic muscarinic receptors. This in turn leads to acetylcholine release and a subsequent increase in lower esophageal sphincter and gastric tone, improved stomach–duodenum coordination, and accelerated gastric emptying [47,49,50,51]. However, these effects are limited to the proximal gut [52].

### 4.6. Other Agents

Prokinetic agents play a crucial role in enhancing gastrointestinal motility by amplifying and coordinating muscular contractions throughout the digestive system, facilitating the movement of contents throughout the gut. While some prokinetics target specific areas of the gastrointestinal tract, others exert more generalized effects based on the distribution of receptor targets for these pharmacological agents. Our study focuses on the utilization of prokinetics in addressing gastric motility disorders, exploring a range of agents, both approved and investigational, that target various receptors and neurotransmitters involved in gastric emptying to exert their prokinetic effects [53].

Similar to metoclopramide, domperidone is another D2 receptor antagonist. Domperidone acts similarly to metoclopramide and is used in gastroparesis. It is provided by the Food and Drug Administration (FDA) as an investigational drug as it is not yet FDA approved [54,55]. Domperidone has been associated with cardiac arrhythmias, which led to its withdrawal from the over-the-counter list in Europe.

Neostigmine is a short-acting agent that is mostly used in a hospital setting due to its short duration of action [56]. It exerts its prokinetic effect via inhibiting acetyl cholinesterase [56]. Pyridostigmine is another agent of a similar mechanism. It has a longer duration of action and is available in a tablet formulation. It has been studied in the pediatric population and was found to have an upper GI promotility effect with symptomatic relief in that population; however, its cardiopulmonary side effects limit its use in the setting of upper G.I. bleeding [57].

Another mechanism of improved gastric emptying is activation of 5HT4 receptors. A few medications were available under that class, including the following: cisapride, clebopride, cinitapride, and mosapride. Cisapride showed improved gastric emptying in early short- and long-term controlled trials [58,59]. However, it was eventually withdrawn from many countries due to significant cardiac arrhythmias associated with its use. The remaining drugs are not easily available, and evidence of their efficacy or safety is scarce [60,61,62]. Clebopride exerts its effect by 5HT4 receptor activation as well as D2 receptor inhibition. Cinitapride is a 5HT1 and 5HT4 agonist with an antagonist effect on 5HT2. Mosapride is strictly a 5HT4 agonist. Novel agents have been developed and are selective agonists for 5HT4 receptors such as prucalopride, velusetrag, naronapride, and felcisetrag. Prucalopride is FDA approved for the treatment of chronic constipation, with studies showing efficacy in symptom relief [63]. Another study showed similar findings with velusetrag [64]. Felcisetrag is better studied than previous drugs. It has been shown to significantly increase gastric emptying, and in the critically ill population, it decreased gastric retention of enteral feeding [65,66]. Velusetrag and felcisetrag had a better safety profile without evidence of cardiac arrythmia [67,68].

Aprepitant is another agent that was found to improve gastroparesis symptoms. It works on D2/D3 receptor and neurokinin 1 (NK) as an antagonist [69]. A novel agent called tradipitant is a NK1 receptor antagonist that significantly improved gastroparesis symptoms in a randomized controlled trial; however, these agents have not been used in the setting of PUD [70].

Relamorelin is a synthetic pentapeptide that acts as a ghrelin receptor agonist. It is proven to accelerate gastric emptying in patients with diabetes and increase the frequency of antral contraction [71,72].

Similar to erythromycin, azithromycin and clarithromycin belong to the class of macrolide antibiotics and exhibit a similar mechanism of action in accelerating gastric emptying through the agonism of motilin receptors. Despite their pharmacological similarity and potential prokinetic effects, neither azithromycin nor clarithromycin have been subjected to comprehensive studies to validate their efficacy or safety profiles in this particular clinical context [73,74]. Furthermore, their utilization introduces an additional layer of concern regarding the development of microbial resistance, a topic that remains the subject of ongoing debate within the medical community. A comparison of all previously discussed medications is shown below in Table 4.

### 4.7. Recommendations

In 2012, the American Society for Gastrointestinal Endoscopy (ASGE) put forth a recommendation hinting at the potential utility of prokinetics prior to endoscopy [2]. However, it is noteworthy that this suggestion was not intended for routine application. Concurrently, within the same year, the ASGE also advocated for the use of pre-endoscopy erythromycin to enhance the diagnostic yield [3]. Nevertheless, the ASGE guidelines elucidated that despite this recommendation, there was no discernible alteration in clinical outcomes attributable to its use [3]. Later, the European Society of Gastrointestinal Endoscopy (ESGE) took a decisive stance in their 2015 guidelines, strongly endorsing the use of pre-endoscopy erythromycin [4]. Our study outcomes with erythromycin use align with those recommendations proving the efficacy of erythromycin compared to a placebo. When compared to metoclopramide, it was also proven to be superior in reducing the need for a second-look EGD.

### 4.8. Clinical Application

Prokinetic agents have gained popularity for use in emergent upper G.I. bleeding. Due to their ability to decrease the gastric transit time, erythromycin has been shown to improve the visualization of the gastric mucosa compared to a placebo and increase the diagnostic yield of endoscopy [75,76,77,78,79,80,81]. A randomized controlled study performed in 2006 found clots in the stomach in 30% of patients receiving pre-endoscopy erythromycin, with 52% of patients in the placebo group having clots within the abdomen [75,76,77,78,79,80,81]. Use of erythromycin to decrease the likelihood of a second endoscopy is particularly interesting. Many published studies have seen a reduced need for a second-look endoscopy with its use [44,45,75,82]. When prokinetic agents such as erythromycin are compared to nasogastric tube or gastric lavage for visualization and the need for repeat endoscopy, results have been mixed in the literature [76,77].

Erythromycin has additionally been shown to decrease the endoscopy duration and reduce the number of units of blood transfused. It may also shorten the overall duration of hospital stay [44,82,83]. Additionally, a study from 2007 investigated cost and noted overall cost savings associated with using erythromycin before EGD [78]. Metoclopramide has yet to be as widely studied, and although it can theoretically improve visualization during EGD, further information is needed [79]. Erythromycin administration pre-endoscopy in patients with peptic ulcer bleeding has been included in the surgical guidelines; however, this recommendation remains weak and is based on moderate-quality evidence [80].

### 4.9. Our Study and Available Data

A recently published double-blinded randomized controlled trial conducted in Thailand and featured in the January 2024 issue of the *American Journal of Gastroenterology* sheds light on the efficacy of metoclopramide in the context of upper gastrointestinal bleeding (UGIB) management [81]. Interestingly, the trial findings revealed that while metoclopramide did not significantly improve visualization during endoscopy, it did exhibit a reduction in the need for a repeat EGD when compared to a placebo. However, a notable limitation of this study was its small sample size, comprising only 31 patients in the metoclopramide group, which poses challenges in generalizing the findings to broader patient populations. In contrast, our study presents compelling evidence suggesting the superiority of erythromycin over metoclopramide in enhancing endoscopic visualization and decreasing the necessity for a second-look EGD. Moreover, our extensive dataset, derived from real-time patient data encompassing a large sample size, enhances the generalizability and reliability of our study findings. These data should further encourage using erythromycin in particular prior to EGD to decrease the need for a second-look EGD.

Additionally, our study mortality rates in both metoclopramide and no medications arms were 1.08%, aligning with the findings of a recently published nationwide study in January 2023 that looked at overall mortality from non-variceal UGIB between the years 2008 and 2018, which found that the overall adjusted mortality ranged from 1.82% to 2.46% [84]. The similar noted mortality rate between the studies further strengthens the value of our data and proves its relevance.

Furthermore, a multicenter, randomized, double-blinded controlled trial that was recently published on 31 March 2024, had similar findings to ours. They found that pre-EGD administration of metoclopramide improves the quality of endoscopic mucosal visualization in patients with variceal bleeding. However, they found no difference in the rate of repeat EGD between the metoclopramide and placebo groups [85].

This further supports our findings and proves that the use of metoclopramide does not reduce the need for repeat EGD in patients with UGIB. Additionally, our study was able to reproduce the superiority of erythromycin use as recommended by the previously mentioned guidelines. This further strengthens the other findings of our study showing no benefit of metoclopramide use prior to EGD, as we have used the same analysis with the same cohorts.

### 4.10. Strengths and Limitations

Our study is characterized by several notable strengths that contribute to its robustness and reliability. Firstly, the considerable size of our patient cohort, coupled with the utilization of a nationwide multi-institutional database, enhances the statistical power of our analysis and facilitates broader generalizability within the United States healthcare context. This extensive database enables us to draw conclusions that are reflective of diverse patient demographics and clinical practices across various regions. Furthermore, the implementation of propensity score matching (PSM) methodology serves as a vital tool in mitigating potential selection biases, thereby ensuring the comparability of patient cohorts across all study arms. This approach minimizes confounding variables and enhances the validity of our findings. Moreover, our study demonstrates balanced representation in terms of gender and race across all cohorts, indicating a true representative sample of the general population. This equitable distribution allows for a comprehensive examination of outcomes without the influence of demographic disparities.

While our study offers valuable insights, it is essential to acknowledge several limitations that may affect the interpretation and applicability of our findings. Firstly, the databases we analyzed primarily consisted of data from American patients, thereby limiting the generalizability of our conclusions to Eastern countries and other regions with differing healthcare systems and patient populations. This geographic restriction underscores the importance of conducting similar studies in diverse global settings to validate our observations and ascertain their broader applicability. Secondly, although we employed validated outcome definitions and propensity score matching to minimize bias, it is important to note that misclassification bias and residual confounding may still be present due to inherent weaknesses within the electronic health records study design. Despite our best efforts to control for these factors, the potential for residual biases cannot be entirely eliminated. Furthermore, the retrospective nature of our study, based on deidentified databases, posed challenges in accessing detailed operative reports from endoscopies. This limitation hindered our ability to thoroughly evaluate the adequacy of initial endoscopic procedures and conduct a comprehensive assessment. Lastly, due to the innate nature of TriNetX research database, we were unable to perform a unified comparison between all three study arms. Future studies incorporating more comprehensive data collection methods, including access to operative reports, could provide a more thorough understanding of the factors influencing patient outcomes in UGIB management.

## 5. Conclusions

This study found that metoclopramide does not decrease the need for a second-look EGD in patients with UGIB when compared to those who do not receive any prokinetic. Metoclopramide was also inferior to erythromycin in its ability to lower the need for a second-look EGD. Therefore, if a prokinetic agent is to be used to improve visualization and decrease the need for a second-look EGD, erythromycin shows superior reduction compared to metoclopramide.

## Figures and Tables

**Figure 1 life-14-00526-f001:**
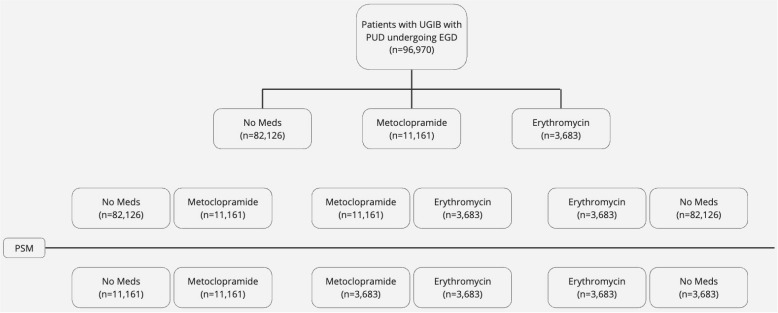
Study flow diagram.

**Figure 2 life-14-00526-f002:**
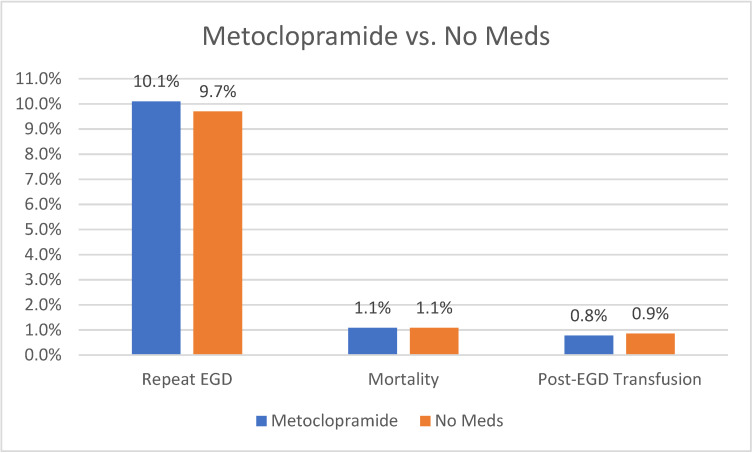
Outcomes of Metoclopramide vs. No Meds.

**Figure 3 life-14-00526-f003:**
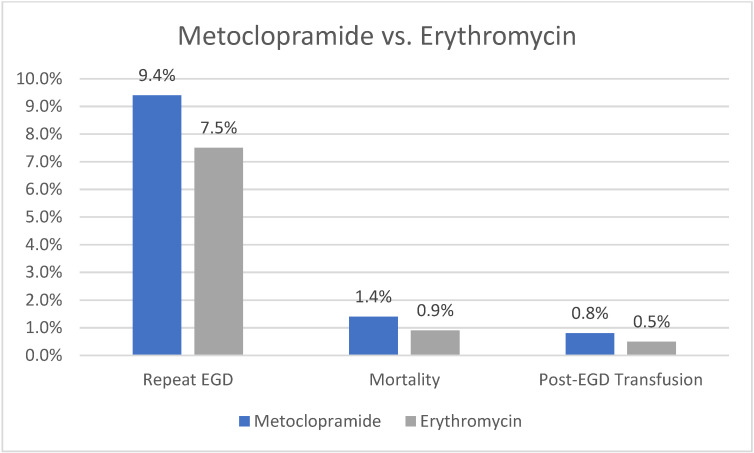
Outcomes of Metoclopramide vs. Erythromycin.

**Figure 4 life-14-00526-f004:**
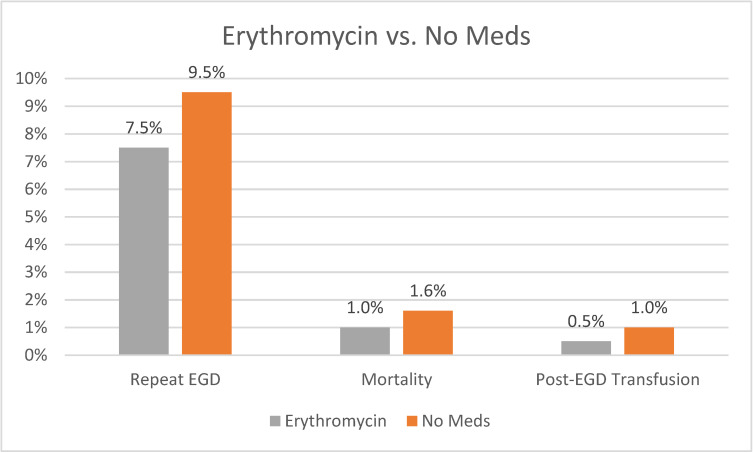
Outcomes of Erythromycin vs. No Meds.

**Figure 5 life-14-00526-f005:**
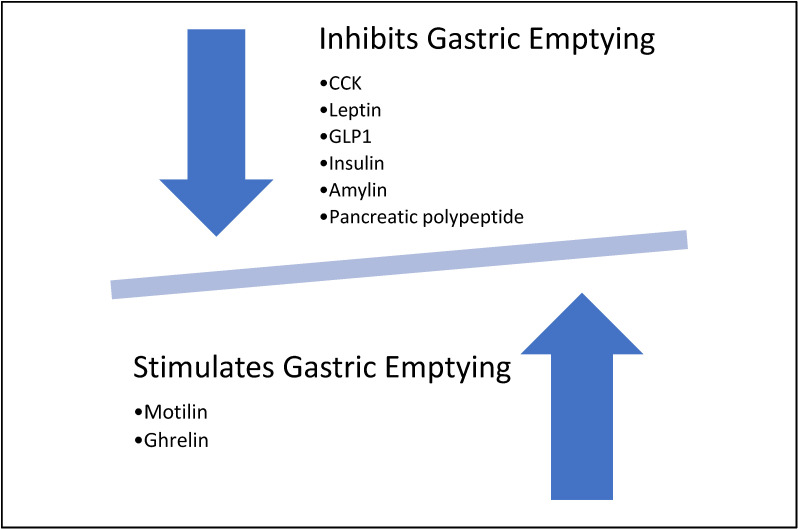
Effects of different neurotransmitters on gastric emptying.

**Table 1 life-14-00526-t001:** Patient Demographics before PSM.

	Before PSM		
	Metoclopramide (*n* = 11,161)	Erythromycin (*n* = 3683)	No Meds (*n* = 82,126)
Age ± SD	56.7 ± 16	64.9 ± 16.5	65.7 ± 15.1
BMI	28.8 ± 6.6	28.7 ± 6.2	28.7 ± 6.1
Female	46.6%	43.4%	39.1%
Not Hispanic or Latino	19.1%	20.3%	15.5%
Hispanic or Latino	2.2%	1.7%	1.5%
White	15.2%	17.4%	13.1%
Asian	0.09%	0.3%	0.2%
Black or African American	4%	3%	2.5%

**Table 2 life-14-00526-t002:** Patient Demographics after PSM.

	Arm 1 after PSM			Arm 2 after PSM			Arm 3 after PSM		
	Metoclopramide (*n* = 11,161)	No Meds (*n* = 11,161)	*p*-Value	Metoclopramide (*n* = 3683)	Erythromycin (*n* = 3683)	*p*-Value	Erythromycin (*n* = 3683)	No Meds (*n* = 3683)	*p*-Value
Age ± SD	56.7 ± 16	56.7 ± 16	0.98	64.4 ± 16.3	64.4 ± 16.4	0.94	64.9 ± 16.5	65 ± 16.5	0.75
BMI	28.8 ± 6.6	29.1 ± 6.4	0.92	28.3 ± 6.3	28.8 ± 6.3	0.42	28.7 ± 6.2	28.7 ± 6	0.97
Female	46.6%	46.6%	0.99	43.4%	43.7%	0.76	43.4%	43.4%	0.98
Not Hispanic or Latino	19.1%	19.2%	0.93	20%	20.2%	0.86	20.5%	20.3%	0.89
Hispanic or Latino	2.2%	2.2%	0.92	1.5%	1.8%	0.31	1.7%	1.7%	1
White	15.2%	15.3%	0.91	16.8%	17%	0.73	17.5%	17.5%	0.98
Asian	0.09%	0.1%	0.82	0.3%	0.3%	1	0.3%	0.3%	1
Black or African American	4%	3.8%	0.53	3%	3%	0.94	3%	2.9%	0.73

**Table 3 life-14-00526-t003:** Summary of Results.

	Metoclopramide (*n* = 11,161)	No Meds (*n* = 11,161)	* p * -Value	Metoclopramide (*n* = 3683)	Erythromycin (*n* = 3683)	* p * -Value	Erythromycin (*n* = 3683)	No Meds (*n* = 3683)	* p * -Value
Repeat EGD	10.1%	9.7%	0.34	9.4%	7.5%	0.003	7.5%	9.5%	0.003
Mortality	1.08%	1.08%	0.95	1.4%	0.9%	0.1	1%	1.6%	0.02
Post EGD Transfusion	0.78%	0.86%	0.5	0.8%	0.5%	0.19	0.5%	1%	0.02

**Table 4 life-14-00526-t004:** Comparison of Prokinetics.

	Receptor	Antiemetic	Gastric Emptying	Comments
Metoclopramide	D2 Antagonist 5HT4 Agonist	Yes	Improve	Extrapyramidal symptoms QT prolongation
Domperidone	D2 Antagonist	Yes	Improve	Investigational
Neostigmine	acetyl cholinesterase Inhibitors	Unknown	Yes	Bradycardia Studies limited to critically ill-patient
Cisapride, cinitapride, and mosapride	5HT4 Agonist 5HT3 Antagonist	Unknown	Improve	QT prolongation
Prucalopride, velusetrag, naronapride, and felcisetrag.	5HT4 Agonist	Unknown	Improve	Being studied
Clebopride	5HT4 Agonist D2 Antagonist	Unknown	Unknown	Not being studied
Tradipitant	D2/D3 Antagonist NK1 Agonist	Unknown	Unknown	
Aprepitant	D2/D3 Antagonist NK1 Agonist	Unknown	No	
Relamorelin	Ghrelin Agonist	Yes	Yes	Being studied
Erythromycin	Motilin Agonist	Unknown	Yes	Risk for tachyphylaxis Risk of arrythmia
Azithromycin	Motilin Agonist	Unknown	Yes	Risk of antimicrobial resistance
Clarithromycin	Motilin Agonist	Unknown	Yes	Risk of antimicrobial resistance

## Data Availability

The available data are presented. Additional data are only available as permitted by a third party.

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
