# Peer review of "The Use of Pre-Endoscopic Metoclopramide Does Not Prevent the Need for Repeat Endoscopy: A U.S. Based Retrospective Cohort Study"

_life, 2024, doi:10.3390/life14040526_

Round 1

Reviewer 1 Report

Comments and Suggestions for Authors

The authors presented an interesting manuscript regarding the use of metoclopramide, erythromycin for evaluating the comparative effectiveness and safety profile of pre-endoscopy administration of these two medications. 

There are a few comments the authors need to address.

The Introduction of the manuscript is too short. Although the descriptions made in the Discussion section are highly appreciated, some brief explanations also need to be made in the introduction about the neurotransmitters and chemicals that are involved in gastric emptying. References need to be added. What is the scope of this study? What is the aim of this research? The authors need to clearly explain these. 

It would be appreciated if the authors also explained why their research and results are better than those of other studies.

Some improved explanations need to be made in 4.9. Our Study and Available Data section. I'm afraid this section is a bit short. 

The conclusions don't reflect the scope, studies conducted and discoveries in this manuscript. The authors need to rewrite their conclusions. 

Comments on the Quality of English Language

Minor editing of English language is required.

Author Response

Thank you so much for your time and your critical feedback. Your comments help us enhance our reporting!

Response to Reviewer 1:

“The authors presented an interesting manuscript regarding the use of metoclopramide and erythromycin for evaluating the comparative effectiveness and safety profile of pre-endoscopy administration of these two medications”

There are a few comments the authors need to address.

  1. The Introduction of the manuscript is too short. Although the descriptions made in the Discussion section are highly appreciated, some brief explanations also need to be made in the introduction about the neurotransmitters and chemicals that are involved in gastric emptying. References need to be added. What is the scope of this study? What is the aim of this research? The authors need to clearly explain these.

Thank you for your feedback! We made the suggested edits; we highlighted the neurotransmitters and chemicals in a quick and succinct way in the introduction with the corresponding references.

The study's scope is focused on evaluating the comparative effectiveness of pre-endoscopic administration of metoclopramide compared to erythromycin in the context of upper gastrointestinal bleeding. We rephrased the introduction to clearly showcase that.

  1. It would be appreciated if the authors also explained why their research and results are better than those of other studies.

Thank you for highlighting that! We highlighted our study strengths in section 4.9. and 4.10.

  1. Some improved explanations need to be made in 4.9. Our Study and Available Data section. I'm afraid this section is a bit short.

Thank you for the feedback. We went ahead and further expanded section 4.9. and added more explanation and recent evidence that was just published last month.

  1. The conclusions don't reflect the scope, studies conducted and discoveries in this manuscript. The authors need to rewrite their conclusions.

Thank you for highlighting this. We rephrased our conclusion to reflect that.

Reviewer 2 Report

Comments and Suggestions for Authors

In this study, the authors aimed to evaluate the comparative effectiveness of pre-endoscopy administration of Metoclopramide against Erythromycin or no medication. Pre-endoscopy administration of prokinetics agents is a common practice and therefore data determining their actual role is very interesting and useful.

A great strength of this study is that it included very big samples to compare, giving high reliability of the results. However, some other issues should be addressed.

First, it is difficult to understand the type of the study. Was it a retrospective analysis or a cohort study? Please report the type of the study more clearly.

A major issue is why the authors did not compare erythromycin versus no medication. In my opinion, this is a crucial omission of the study. The authors had the data and they should have made the same comparisons between erythromycin and no-meds groups.

Moreover, regarding the description of the sample, somatometric parameters such as BMI would be interesting to be presented among the other baseline characteristics.

In addition, how do the authors explain the statistically significant differences regarding gender, age, and race between the three groups before PSM?

In line 234, the authors referred to their study as a review. If so, then the execution of the review should be described(which databases were assessed, which keywords were used, when it was held, etc.)

In the discussion, the authors should focus on the available data on the field of pre-endoscopy administration of prokinetics agents and compare their results to other studies' results rather than providing so much information regarding the physiology which is common knowledge.

To sum up, this is a valuable study in general and by improving its content based on the aforementioned it could be considered suitable for publication by your esteemed journal.

Author Response

Thank you so much for your critical feedback, time, and effort! With your help, we were able to enhance our manuscript.

Response to Reviewer 2:  

“In this study, the authors aimed to evaluate the comparative effectiveness of pre-endoscopy administration of Metoclopramide against Erythromycin or no medication. Pre-endoscopy administration of prokinetics agents is a common practice and therefore data determining their actual role is very interesting and useful”

A great strength of this study is that it included very big samples to compare, giving high reliability of the results. However, some other issues should be addressed.

  1. First, it is difficult to understand the type of the study. Was it a retrospective analysis or a cohort study? Please report the type of the study more clearly.

Thanks for highlighting that. This is a retrospective cohort study; we will rewrite the title to reflect so.

  1. A major issue is why the authors did not compare erythromycin versus no medication. In my opinion, this is a crucial omission of the study. The authors had the data and they should have made the same comparisons between erythromycin and no-meds groups.

Thank you for highlighting this. We have the data indeed; we did not want to shed the light away from the metoclopramide findings since the erythromycin was well-reported and addressed by ASGE. We went ahead and added the metoclopramide vs no meds to the results section and table and further addressed it in section 4.9.

  1. Moreover, regarding the description of the sample, somatometric parameters such as BMI would be interesting to be presented among the other baseline characteristics.

This is an excellent point that should have added! We went ahead and added the BMI of each cohort.

  1. In addition, how do the authors explain the statistically significant differences regarding gender, age, and race between the three groups before PSM?

Great question. Prior to PSM there is a significant discrepancy between the 3 cohorts in terms of comorbidities and characteristics. This is in large secondary to the great discrepancy in the cohorts’ patient count. The most important element is making sure there is no statistically significant difference between all cohorts in every study arm to ensure comparability.

  1. In line 234, the authors referred to their study as a review. If so, then the execution of the review should be described (which databases were assessed, which keywords were used, when it was held, etc.)

Thanks for pointing that out! We rephrased the sentence.

  1. In the discussion, the authors should focus on the available data on the field of pre-endoscopy administration of prokinetics agents and compare their results to other studies' results rather than providing so much information regarding the physiology which is common knowledge.

Thanks for the feedback. We implemented those changes by adding more content to section 4.9. to compare other studies with ours and more recent evidence that was published within the last 30 days.

  1. To sum up, this is a valuable study in general and by improving its content based on the aforementioned it could be considered suitable for publication by your esteemed journal.

Thank you so much for your time and feedback. Your input helps us strengthen our content and further improve our reporting.

Round 2

Reviewer 1 Report

Comments and Suggestions for Authors

The authors revised the manuscript and implemented the suggestions.

Author Response

Thank you for your time and feedback which helped us improve our manuscript.

We further adjusted the conclusion to appropriately reflect our study findings.

Reviewer 2 Report

Comments and Suggestions for Authors

All the changes made by the authors contributed to improving the content of their study. 

However, there are some issues.

First, in Table 1: the p-value (p<0.0001) regarding BMI according to my calculation based on the provided summary data is incorrect. It is estimated to be p=0.272 (Tukey HSD Post-hoc Test:

Group 1 vs Group 2: Diff=-0.1000, 95%CI=-0.3744 to 0.1744, p=0.6695

Group 1 vs Group 3: Diff=-0.1000, 95%CI=-0.2457 to 0.0457, p=0.2420

Group 2 vs Group 3: Diff=0.0000, 95%CI=-0.2432 to 0.2432, p=0.9948).

Please re-check the p-value regarding BMI.

Second, based on the fact that the comparison was made between 3 independent groups, the proper test for this purpose is ANOVA (when normality is assumed) or Kruskal-Wallis test (when the data are skewed) and then post hoc test (such as Tukey) for identifying sub-groups differences. The authors performed an independent t-test comparing two out of three groups every time. This increases the possibility of Type II error due to multiple comparisons. So after adding the third type of comparison between Erythromycin and No-Meds, the authors should have performed ANOVA or Kruskall-Wallis and then completed the comparisons with post-hoc subgroup analysis. I believe that the same has been performed when comparing the continuous baseline characteristics.

Please, make sure that the proper statistical methodology was followed and in my opinion, this study would be precious to be published after the methodology is warranted to be the suitable one.

Author Response

Thank you again for your constructive feedback to ensure the quality of the published manuscript and material!

I just wanted to highlight that we did not compare all three study arms. We intended to compare metoclopramide to no meds and then to erythromycin. The third arm of erythromycin compared to no meds proves what is already known in the literature and ensures our accurate statistical methodology as well as inclusion and exclusion criteria, however, it is not intended to be used to draw conclusions. That being said, what you suggest would indeed minimize errors and provide a unified analysis and conclusion. Our inability to do so is a limitation of our study due to limited resources as well as the fact that our TriNetX research database does not allow for more than one study arm at a time. We understand that this is a limitation of our study and we included it into section 4.10. Additionally, to avoid confusion of readers or the assumption that we compared all 3 study arms, we removed the p-value columns from Table 1 leaving just the description of each cohort. We are also highlighting the p-values post-PSM which is the most important part of the analysis that ensures comparability of the groups.

Thank you again for your time and effort!

Round 3

Reviewer 2 Report

Comments and Suggestions for Authors

Responding immediately, the authors explained and highlighted in the manuscript their methodology efficiently. I appreciate their effort, and I believe that this study is worth to be published in your esteemed journal.